# The Trophic Niche of Two Sympatric Species of Salamanders (Plethodontidae and Salamandridae) from Italy

**DOI:** 10.3390/ani12172221

**Published:** 2022-08-29

**Authors:** Enrico Lunghi, Claudia Corti, Marta Biaggini, Yahui Zhao, Fabio Cianferoni

**Affiliations:** 1Key Laboratory of the Zoological Systematics and Evolution, Institute of Zoology, Chinese Academy of Sciences, Beijing 100101, China; 2Division of Molecular Biology, Institut Ruđer Bošković, 10000 Zagreb, Croatia; 3Natural Oasis, 59100 Prato, Italy; 4Unione Speleologica Calenzano, 50041 Florence, Italy; 5Museo di Storia Naturale dell’Università degli Studi di Firenze, Museo “La Specola”, 50125 Florence, Italy; 6Istituto di Ricerca sugli Ecosistemi Terrestri (IRET), Consiglio Nazionale delle Ricerche (CNR), 50019 Florence, Italy

**Keywords:** *Speleomantes*, *Hydromantes*, *Salamandra*, diet, forest, competition, prey selection

## Abstract

**Simple Summary:**

Studies on species’ trophic niches are essential to understand the characteristics of species’ ecology and life traits, as well as to improve conservation strategies. In the absence of competitors, species realize their trophic niche including in their diet the most profitable food resources. In the presence of competitors, species modify their preferences to reduce competition and maintain the highest benefits at the same time. In this study, we assessed the trophic niche of two species of salamanders coexisting in a forested area of Italy and evaluated which might be the mechanisms that these two species adopted to reduce competition. We found that the Italian cave salamander (*Speleomantes italicus*) mostly consumed flying prey with a hard cuticle, while the fire salamander (*Salamandra salamandra*) preferred worm-like and soft-bodied prey. In conclusion, we hypothesize that in our case, the two species of salamanders did not have to change their prey preference in order to avoid competition, but divergences in metabolism and behavioral traits likely worked as natural deterrent.

**Abstract:**

The trophic niche of a species is one of the fundamental traits of species biology. The ideal trophic niche of a species is realized in the absence of interspecific competition, targeting the most profitable and easy-to-handle food resources. However, when a competitor is present, species adopt different strategies to reduce competition and promote coexistence. In this study, we assessed the potential mechanisms that allow the coexistence of two generalist salamanders: the Italian cave salamander (*Speleomantes italicus*) and the fire salamander (*Salamandra salamandra*). We surveyed, in April 2021, a forested area of Emilia-Romagna (Italy) during rainy nights. Analyzing the stomach contents of the captured individuals, we obtained information on the trophic niche of these two sympatric populations. Comparing our results with those of previous studies, we found that the two species did not modify their trophic niche, but that alternative mechanisms allowed their coexistence. Specifically, different prey preferences and predator metabolisms were likely the major factors allowing reduced competition between these two generalist predators.

## 1. Introduction

The trophic niche is one of the fundamental traits of species biology [1]. The study of the trophic niche provides important information on multiple species traits such as behavior (e.g., foraging strategy and prey selection), physiology (e.g., specific nutritional requirements and metabolism), and its trophic position in the local community [2,3,4,5]. Nonetheless, this type of study may be pivotal to implement conservation strategies for species that need particular protection [6]. In the absence of heterospecific competitors, a species tends to develop its trophic niche by targeting the most profitable resources in terms of nutritional intake and handling ability [4,7]. Under this condition, the overall trophic spectrum of the population can mirror the shared preference of individuals (when they exploit the same typologies of food resources), or it can result from the combination of different preferences when intraspecific competition forces individuals to target only a subset of the food resources consumed by the entire population [8,9]. When co-occurring species compete for the same food resources, their realized trophic niche usually differs (at least for the weaker competitor) from the ideal one, as individuals have to switch to alternative resources to reduce the competition and be able to coexist [10,11].

We here assessed for the first time the trophic niche of two sympatric generalist salamanders: the Italian cave salamander *Speleomantes italicus* and the fire salamander *Salamandra salamandra*. The trophic niche of these two species has only been assessed in the absence of potential competitors e.g. [12,13,14], leaving unknown the mechanisms and the extent to which these species are able to modify their trophic niche to coexist with competitors. *Speleomantes italicus* is one of the eight species of Plethodontidae occurring in Europe, and its distribution encompasses the Apennine chain, from northern Tuscany to Abruzzo (Italy), where it occurs in forested areas and subterranean environments as well [15,16]. *Speleomantes* are lungless salamanders that require specific microhabitat conditions (i.e., high humidity and relatively low temperature) to maintain the high efficiency of cutaneous respiration [17,18]. *Speleomantes* are fully terrestrial amphibians that live and reproduce exclusively in subaerial environments [15]. Courtship can occur all year round, while gravid females lay their eggs twice per year (beginning of spring or autumn) in hidden places where they provide prolonged parental care (≥4 months) until the hatchlings are ready to leave the nest [19,20,21,22]. The narrow microhabitat requirements and the *k*-selected reproductive strategy of these species make them deserving of special protection [18,23,24]. Although being epigean species, *Speleomantes* gained the vernacular name of “cave salamanders” because they can be easily observed in natural and artificial subterranean environments [15,25]. These species are therefore able to prey in both surface and subterranean environments [26,27,28,29]. The trophic niche of *S. italicus* was only studied in subterranean populations [12]. Researchers have observed high variability in the consumed prey among populations, with a clear predominance of Diptera, one of the most abundant prey in subterranean environments [30,31].

*Salamandra salamandra* is the most widespread species of Salamandridae in Europe, ranging from the Iberian Peninsula to the western part of Ukraine, including the Balkans and all central European countries [32]. Despite being a widely spread Urodela, the fire salamander recently faced a huge decline in some parts of its distribution due to infection with the fungus *Batrachochytrium salamandrivorans* [33]. The fire salamander is a typical biphasic amphibian that has aquatic larvae and terrestrial adults [34]. Courtship occurs from spring to autumn, and females usually maintain the fertilized eggs in their bodies until the following spring. When eggs are ready, the female enters into a body of water to release the newly hatched larvae, which need at least 6 months to metamorphose into the adult form [34]. Although *S. salamandra* usually reproduces in surface water, several cases of reproduction in subterranean environments are also known [35,36]. The larvae feed upon multiple aquatic invertebrate species, but they can also adopt cannibalism when prey are scarce [14,37,38]. Adults forage in terrestrial environments and they often prey upon “worm-like” prey such as annelids, diplopods (among arthropods), and “slugs”, defined as apparently shell-less terrestrial gastropods (among mollusks) [13,39,40].

The present study aimed to assess whether *S. italicus* and *S. salamandra* may be potential competitors when they occur in sympatry and to determine the mechanisms preventing the competition. In order to coexist, species tend to reduce the competition by targeting different types of prey [41,42]; therefore, we expected a little overlap in the realized trophic niche of these populations when occurring in sympatry. In addition, we evaluated potential divergences in foraging behavior among conspecific individuals.

## 2. Materials and Methods

We carried out surveys in a forested area of the northern Apennines in the province of Bologna (Emilia-Romagna, Italy); precise information on the site is omitted to ensure species protection [43]. The surveys (six in total) were carried out in April on rainy nights (7 p.m.–2 a.m.) with random frequency, but with at least a one-day interval between two surveys. We surveyed an area of about 4000 m^2^ in search of individuals of *Speleomantes italicus* and *Salamandra salamandra* on the ground, on trees, and on dry stone walls [44]. The captured salamanders were photographed using a portable photo-studio [45]. We estimated the snout-vent length (SVL, in mm) of salamanders from the images using ImageJ software [46,47,48]. We focused only on SVL as this measure provides more accurate information on both the age and size of the salamanders [23], as the tail can be lost (and possibly regenerate) due to predation events [49]. The images were also used to individually recognize the salamanders of both species by observing their dorsal pattern [50,51]; for *S. italicus*, an additional marking method was also employed (i.e., visual implant elastomers) [52]. Individuals of both species were sexed based on their size and the presence of distinctive sexual characters. For *S. italicus*, we used the size of 50 mm (SVL) as a threshold to distinguish adults (≥) and juveniles (<) [23]. Among adults, male recognition was based on the presence of their typical secondary sexual characters (i.e., mental gland, prominent pre-maxillary teeth, and conical shape of the head) [15]; all adult individuals lacking these traits were considered females. In *S. salamandra*, we used the size of 90 mm (SVL) to distinguish between adults (≥) and juveniles (<). Among adults, those with swelling at the base of the cloaca were considered to be males [49]. We weighed all the salamanders using a digital scale (accuracy 0.01 g) and then inspected their stomach residues by stomach flushing, a harmless technique that allows to obtain information on the individual’s latest foraging activity [26,53]. Prey residues were recognized at the order level, and each order represents a single prey category [26]. In some cases, further distinctions were also made at the family level, or between larval stages, i.e., when these groups are morphologically distinct and characterized by different ecology (e.g., aquatic vs. subaerial) (see Table 1). Each of these groups was considered an independent category. For additional information on the method, see [26,44].

We used the analysis of similarity (ANOSIM with 10,000 permutations) to assess whether the similarity in the trophic niche between the two populations was higher than that occurring within each population [54,55]. Furthermore, we tested whether the two populations diverged in terms of multivariate dispersion of their diet (*betadispr* function, 999 permutations) [56]. We used PERMANOVA to assess potential interspecific differences in diet composition [57,58]. The nonmetric multidimensional scaling (NMDS) plot with Euclidean distance was used to show the trophic niche differences between the two populations of salamanders. Although the dataset contained data on individuals captured multiple times (see Results), we decided to maintain all the data in this analysis to have a more complete information on the species’ overall trophic niche.

Generalized linear mixed models (GLMMs) [59,60] were used to assess whether differences between sex and ontogenetic stage in terms of the number and diversity of prey consumed existed. In the first GLMM, the log-transformed number of consumed prey by each individual was used as the dependent variable, while individual SVL and sex/stage (male, female, or juvenile; hereafter only referred as “sex” for the sake of brevity) were used as independent variables; the day of the survey was used as the random factor. In the second GLMM, we used the Shannon index of the prey consumed by each individual as the dependent variable; all the other variables remained the same. The analysis of the two GLMMs were separately performed for *S. italicus* and *S. salamandra*. To avoid bias due to pseudoreplication, we purged the datasets used in GLMM analyses removing the data on recaptured individuals and maintaining only the event in which the number of recognized prey was the largest. We decided not to analyze individuals with empty stomachs or with unrecognized prey, as they represented a very small percentage of the overall pool of individuals (about 6% for *S. italicus* and 3% for *S. salamandra*; see Results).

Finally, we estimated the degree of individual diet specialization that occurred in these two sympatric populations [61]. For each species, we calculated the index of individual specialization (IS) as shown in [62,63]. We focused only on IS, as this index is significantly correlated to the other diet specialization niche metrics [64]. To improve the clarity, we used the index V =1—IS proposed in Bolnick, et al. [65], where values tending to 1 indicate a high degree of individual diet specialization, while values tending to 0 indicate that the population is mostly made up of generalists [62]. Bootstrapping (repeated 9999 times) was used to test whether the observed index of individual diet specialization significantly diverged from the simulated one, i.e., a scenario in which all individuals randomly choose their prey. In this analysis, we further purged the dataset used in GLMM, removing individuals from which we recognized <3 prey items; this was a precaution to not overinflate the individual diet specialization index.

The data on *S. italicus* used here were retrieved from [44], while the data on *S. salamandra* are provided as Appendix A.

## 3. Results

We captured and inspected the stomach contents from 315 *Speleomantes italicus* (128 females, 146 males, and 41 juveniles) (average captured individuals ± SD per night; 52 ± 20.5) and 32 *Salamandra salamandra* (22 females, 9 males, and 1 juvenile) (5 ± 6). Individuals of *S. italicus* were captured during all six surveys, while individuals of *S. salamandra* were only captured in three. Some of them (30 *S. italicus* and 4 *S. salamandra*) were recaptured several times. In the stomachs of most of the captured salamanders, we found residuals of consumed prey; only 18 *S. italicus* had an empty stomach. In two *S. italicus* and one *S. salamandra*, the stomach contents were in an advanced state of digestion, so we were unable to recognize the prey consumed at the established taxonomic level. We recognized 2,900 prey items from *S. italicus* (average ± SD per individual; 9.83 ± 6.56) and 168 from *S. salamandra* (5.42 ± 3.08), belonging to 35 groups of prey (Table 1).

The trophic niche of *S. italicus* included all the prey categories described in Table 1, where just four (Araneae, Entomobryomorpha, Coleoptera, and Diptera) accounted for 51.86% of the consumed prey. The trophic niche of *S. salamandra* included only 15 of the prey categories, with 3 of them (Gastropoda, Diptera-larvae, and Haplotaxida) accounting for 63.1% of the consumed prey. The analysis of similarity identified a significant divergence between the trophic niche of *S. italicus* and that of *S. salamandra* (R = 0.476, *p* = 0.001) (Figure 1A); the diet of the two populations showed a significant heterogeneity of multivariate dispersion (permutation test: *p* = 0.001). The analysis of PERMANOVA confirmed the divergence of the trophic niche between the two populations (*R^2^* = 0.07, *p* = 0.001). The trophic niche of *S. italicus* was much larger than that of *S. salamandra* (Figure 1B).

In *S. italicus*, neither the SVL (*F_1,262.13_* = 1.13, *p* = 0.289) nor the sex (*F_2,260.75_* = 1.35, *p* = 0.261) significantly affected the number of prey consumed. Similar results were obtained for the diversity of prey consumed (SVL: *F_1,261.43_* = 0.1, *p* = 0.748; sex: *F_2,260.19_* = 0.95, *p* = 0.387). In *S. salamandra*, none of the variables affected the number (SVL: *F_1,22.52_* = 0.03, *p* = 0.863; sex: *F_2,22.99_* = 0.6, *p* = 0.558) or diversity of the prey consumed (SVL: *F_1,23_* = 0.26, *p* = 0.612; sex: *F_2,23_* = 0.07, *p* = 0.933). In *S. italicus*, we found a significantly high proportion of specialized individuals (V = 0.621, *p* < 0.001), while in *S. salamandra*, we observed a similar proportion of both generalist and specialist individuals (V = 0.506, *p* = 0.019).

## 4. Discussion

The trophic niche of the plethodontid salamander, *Speleomantes italicus*, was the widest, including in its diet all the prey categories recognized in this study (Table 1, Figure 1B). These results are in line with what has been observed in other populations of *S. italicus* and for the entire genus, as *Speleomantes* generally show a wide variability in the prey consumed regardless of sex and age [12,26,27,28,44,67]. In this epigean population, most of the prey consumed are flying prey (about 43%), similar to what has been observed in subterranean populations of *Speleomantes* [4,35,68]. It may be possible that *Speleomantes*’ protrusible tongue increases their ability to capture flying prey, a skill that has proven to be extremely useful when individuals cling to vertical surfaces [69,70,71]. Conversely, *Speleomantes* are not very attracted by slow wormlike prey [71,72], so the poor representation of these types of prey in their diet may be the result of individual choice [8,64]. Indeed, the overall worm-like prey consumed by this population of *S. italicus* did not even cover 14% of its diet (Table 1). Soft-bodied prey was usually underrepresented in previous studies on *Speleomantes*’ diet, as this type of prey is likely rapidly digested without a trace [12,26,27,66]. In the studied epigean population, this type of prey (i.e., Pulmonata and Haplotaxida) represented approximately 2.5% of the prey consumed (Table 1). Notably, not only snails were consumed in this population, but also slugs, a type of prey never reported for *Speleomantes* before. It could be possible that, in our case, individuals underwent stomach flushing shortly after ingesting the prey, without having completed their digestion. The stomach contents of the subterranean *Speleomantes* populations were usually obtained during the day (9 a.m.–6 p.m.) [26,44]. *Speleomantes* come out of their subterranean shelter to feed on the surface, especially on cold and humid nights [15,73]; therefore, checking the stomach contents the day after foraging may be too late to obtain information on prey being quickly digested. No significant effect on the number of consumed prey was found in this population of *S. italicus*. In a previous study, it was observed that juvenile *Speleomantes* from subterranean populations consumed significantly less prey than adults [4]. In this study, we surveyed a fully epigean population, meaning that individuals only forage in the surface environment, where the diversity and availability of prey are remarkably different from that found in subterranean environments [74]. A large number of prey consumed in this study were Collembola (Table 1), animals of millimetric size that can be eaten in large number also by juveniles. These prey were widely underrepresented in the stomach contents from subterranean populations [4], despite their steady presence in subterranean environments [75]. This divergence could be due to a variability in the abundance of Collembola between subterranean and surface environments rather than different prey preferences between juveniles inhabiting different environments. This hypothesis remains to be tested.

The trophic niche of *Salamandra salamandra* was narrower, as it included less than half of the overall prey consumed by *S. italicus* (Table 1, Figure 1B). *Salamandra salamandra* behaved exactly the opposite from *Speleomantes*, preferring mostly slow-moving worm-like prey [40,76]. This type of prey consumed by *S. salamandra* accounted for 79% of its diet (Table 1), corroborating similar results obtained from different populations throughout its distribution [13,39,53,77,78]. Among the stomach contents recognized from *S. salamandra*, we observed a very low frequency of prey with hard cuticles, and generally they were <1 cm in size (Table 1). In a recent study, it was shown that soft-bodied prey can be underrepresented when analyzing the species’ diets through stomach flushing [13]. In our study, we observed that most (>63%) of the prey consumed by *S. salamandra* exclusively consisted of soft-bodied prey (Table 1), which means that these taxa are not necessarily underrepresented in this type of study. As we discussed for *S. italicus* (see above), this inconsistency may have been due to the delay between the foraging event and the inspection of salamanders’ stomachs. In their study, Marques, Mata and Velo-Antón [13] investigated the trophic niche of *S. salamandra*, analyzing their feces. This means that the prey had undergone the entire digestive process and that the salamanders had assimilated most of the wet mass by defecating only the solid residues that they had not been able to digest. In this circumstance, it is understandable how soft-bodied prey can only be detected through DNA analysis.

Our hypothesis in which we predicted limited overlap of the trophic niche between the two syntopic species was supported by our results, as only a small portion of the 95% CI of the species trophic spectrum overlaps (Figure 1B). The divergence in prey preference, as well as their size, may play a fundamental role in reducing the competition between these two species, and those preferences may be related to different energy requirements. Lungless and relatively small salamanders such as *S. italicus* do not need a large energy supply because they have a low-energy and efficient metabolism, while larger salamanders with lungs require much more energy to maintain their relatively higher metabolism [79,80]. Soft-bodied prey is mostly composed of moist mass that can be fully digested, providing much more energy (kilojoules) than those that have hard parts that cannot be digested [79]. This may be one of the reasons why *S. salamandra* more frequently consumes soft-bodied prey. This also supports the hypothesis that *S. italicus* rather chooses to not prey on this type of prey, as it does not seem that the surveyed area was characterized by a particularly low abundance of worm-like prey. Although we lack specific data to show, Pulmonata and Haplotaxida consumed by *S. salamandra* had an impressive size, certainly unsuitable for *S. italicus*. A study focusing on six species of *Speleomantes* showed that the size of prey consumed can vary between species and between individuals of different sizes [4]. Therefore, although the overall categories of prey consumed by *S. salamandra* can also be consumed by *S. italicus* (Figure 1B), prey size can be a powerful mechanism to reduce competition. This is a hypothesis we would like to test in the near future. 

In our study the analyzed dataset was quite unbalanced; the data for *Speleomantes italicus* was almost 10 times higher than that of *Salamandra salamandra*. This asymmetry may have affected the robustness of some analyses, so we recommend a careful interpretation of our results.

The two species also diverged in terms of proportion of specialized individuals: the population of *S. italicus* had a higher proportion of specialized individuals, while in *S. salamandra* the proportion of both generalists and specialist individuals were similar. Considering the co-occurrence of these two populations, we can exclude the potential effects of the environment on the frequency of specialized individuals [64]. Alternatively, intraspecific competition may play a major role here [8]. In our study, the observed density of *S. italicus* (i.e., the overall captured individuals) was about 0.07 individuals/m^2^, while for *S. salamandra*, it was ten folds lower (0.007 individuals/m^2^). Previous studies on *Speleomantes* did not provide straightforward information about the potential occurrence of intraspecific competition or the related effects on the diet specialization of individuals [64,81,82,83]. To the best of our knowledge, we are not aware of similar studies performed on adult individuals of *S. salamandra*. Therefore, this remains an open hypothesis that deserves to be tested.

## 5. Conclusions

Our study provides indications on the potential mechanisms used by two co-occurring generalist salamanders to avoid competition. In our case, the two generalist salamanders, although able to target similar prey, mostly chose different prey types, and when they consumed similar prey, they did target prey of different size. *Speleomantes italicus* mostly consumed flying prey with hard cuticle, while *Salamandra salamandra* did the opposite, more frequently consuming worm-like soft-bodied prey. The prey consumed by both species mostly differed in size, making morphological constraints a potential tool useful for reducing competition. In conclusion, morphological constraints, together with other characteristics such as species metabolism (i.e., digestion ability) and prey preference, might play an important role in reducing the competition between sympatric generalist predators.

## Figures and Tables

**Figure 1 animals-12-02221-f001:**
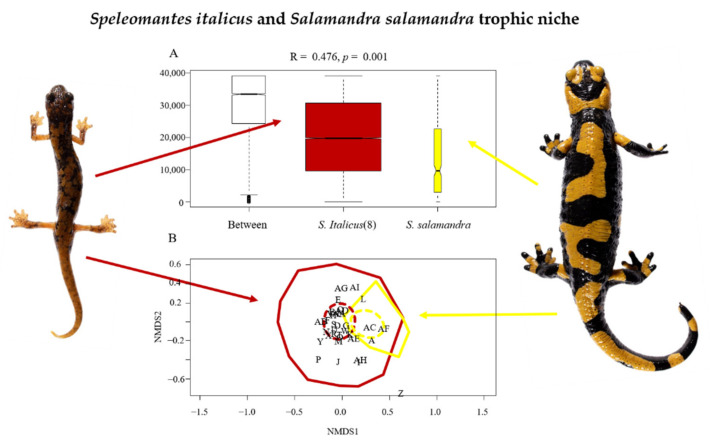
(**A**) Box whisker plot of ANOSIM analysis comparing the diets of *Speleomantes italicus* and *Salamandra salamandra*. Boxes indicate values from 25th (bottom) to 75th (top) percentile; horizontal black line indicates the median; box width is proportional to sample size. (**B**) Cumulative NMDS and dashed 95% confidence ellipses with relative position of each species. NMDS plot needs to be carefully interpreted due to the unbalanced datasets (stress = 0.26). The number (8) indicates the population code for *S. italicus*, which is aligned with that in [44,66]. The two sample animals (*S. italicus* on the left and *S. salamandra* on the right) are not to scale.

**Table 1 animals-12-02221-t001:** List of the prey items found in the stomach contents of *Speleomantes italicus* and *Salamandra salamandra*. To each group of prey, we assigned a code (first column), which we used in the NMDS plot to increase its clarity. In brackets is the relative importance (%) of each group of prey within the trophic niche of each species.

Prey Code	Prey Order	Number Recognized in *Speleomantes italicus* and Relative Importance (%)	Number Recognized in *Salamandra salamandra* and Relative Importance (%)
A	Pulmonata	54 (1.86)	59 (35.12)
B	Sarcoptiformes	78 (2.69)	0
C	Mesostigmata	15 (0.52)	0
S	Trombidiformes	7 (0.24)	0
E	Araneae	359 (12.38)	9 (5.36)
F	Pseudoscorpiones	125 (4.31)	0
G	Opiliones	30 (1.03)	8 (4.76)
H	Lithobiomorpha	22 (0.76)	2 (1.19)
I	Geophilomorpha	14 (0.48)	0
J	Scolopendromorpha	5 (0.17)	0
K	Julida	16 (0.55)	7 (4.17)
L	Polydesmida	92 (3.17)	13 (7.74)
M	Isopoda	81 (2.79)	2 (1.19)
N	Symphypleona	11 (0.38)	0
O	Poduromorpha	35 (1.21)	0
P	Entomobryomorpha	288 (9.93)	0
Q	Blattodea	4 (0.14)	0
R	Hemiptera	186 (6.41)	0
S	Hymenoptera	22 (0.76)	0
T	Hymenoptera-Formicidae	121 (4.17)	1 (0.6)
U	Coleoptera	275 (9.48)	2 (1.19)
V	Coleoptera-Staphylinidae	82 (2.83)	0
W	Coleoptera-larvae	35 (1.21)	4 (2.38)
X	Trichoptera-larvae	3 (0.10)	0
Y	Plecoptera	179 (6.17)	3 (1.79)
Z	Lepidoptera	1 (0.03)	1 (0.6)
AA	Lepidoptera-larvae	24 (0.83)	0
AB	Diptera	582 (20.07)	10 (5.95)
AC	Diptera-larvae	109 (3.76)	23 (13.69)
AD	Archaeognatha	10 (0.34)	0
AE	Speleomantes-skin	5 (0.17)	0
AF	Haplotaxida	22 (0.76)	24 (14.29)
AG	Siphonaptera	3 (0.10)	0
AH	Dermaptera	4 (0.14)	0
AI	Ixodida	1 (0.03)	0

## Data Availability

Data for *Speleomantes italicus* were retrieved from [44], while data for *Salamandra salamandra* are provided as Appendix A.

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
