# Peer review of "The Trophic Niche of Two Sympatric Species of Salamanders (Plethodontidae and Salamandridae) from Italy"

_animals, 2022, doi:10.3390/ani12172221_

Round 1

Reviewer 1 Report

Nice paper, well written.

I do have a bit of a problem with the self citation, it is not inappropriate as far as I can judge as I did not check them all, but I see a lot of Lunghi and very little Salvidio (who has published also many papers on food/ in Speleomantes) in the literature list. Please check this.

Is there a reason not to include the data of Speleomantes in the appendix? You state in line 291 state that there are specialized individuals but not significant; I can not check this now without the data.

Lin 298 you use m2 for densities, would hectares not be more appropriate?

There is a Romanian studies which might be interesting to include? I just add it as pdf.

Reviewer 2 Report

The paper is interesting and overall clear and well written but there is room for some improvements. 

The most important is to take  into  account the huge difference in sample size between the studied populations of the two species (i.e. aprox 10 times more captured individuals of S. italicus than of S. salamandra) when performing the ordination and GLMM. 

I am not sure what is the best way for this. Mantel test is considered less sensitive to unbalanced sampling designs than ANOSIM. However both Mantel and ANOSIM tests were found to not behave reliably for unbalanced designs (Anderson, M. J., & Walsh, D. C. (2013). PERMANOVA, ANOSIM, and the Mantel test in the face of heterogeneous dispersions: what null hypothesis are you testing?. Ecological monographs, 83(4), 557-574). Therefore, search carefully in the literature for any multivariate test that is robust to unbalanced sampling size and if nothing available, do state and discuss this shortcoming of the study. The new analysis may clarify the results. As they are now the results of the multivariate tests and the NMDS indicate different things. ANOSIM and PERMANOVA suggests difference in diet composition between the species while the NMDS indicates that the diet of S. Salamandra overlaps with that of S. italicus. 

Unequal sample size can bias the diversity indices, including Shanon diversity index. Rarefaction  may induce more bias so my advice would be to use estimates of alpha diversity that account for unobserved species. 

Minor comments: 

Line 145. Probably use stage/sex because this explanatory variable is not female vs male but includes also juvenile category. I have never known what is the best way to include this kind of variables in the analysis, either as two variables: sex (female vs male) and stage (adult vs juvenile) or one with three categories as the authors did. 

Table 1. Define N. I guess is the number of preys. Probably using a more intuitive codification of the prey would help the readability of the Figure 2B. For example using the first 4 letters of the Order and the first two letters of the Family or stage. I did not find the Figure 1. It seems that is Figure 2. 

Fig 1 A and 1 B can be improved. Use las =1, to flip the numbers on the y axis. There is a mistake in the legend. Replace _ with . and draw properly the 95% confidence ellipses.  Add the stress value in the Fig caption.

Line 215. It is not clear how the prey categories were defined. From the first phase I get the felling that the number of categories were set a priori. 

Line 250 S. salamandra 

Reading the discussion, as the authors admit, the study can benefit from an analysis of prey size differences in the trophic niche. The size of each item can be probably estimated or taken from the literature, as mean body size of the taxon. After this analysis one of the conclusion that the two salamander species are feed on different prey of different size will be supported, which is not the case in this version of the ms. 

Reviewer 3 Report

The manuscript presents new data on the trophic niches of two sympatric salamanders in Italy. The ecology of both species is well briefly described and authors emphasized the few studies on their diet showing that both when adult are generalist consuming various terrestrial Invertebrates, with however a clear predominance for certain preys (Diptera for S. italicus and worm-like for S. salamandra), but with high local variations. Authors suspected then that when the two species are in sympatry they can enter in competition and they expected differences in the selected preys. However, this study shows that the target preys by S. salamandra are totally included in spectrum of preys consumed by S. italicus which appears to be more generalist. About this, I think that authors should moderate their conclusions considering  that the high difference in the sample size between S. italicus (N=315, number of preys=2900) and S. salamandra (N=32, number of preys= 168) could have bias the results. Indeed, several taxa found in some stomachs of S. italicus are really few frequent. Could we exclude thus to find also them with more sampled S. salamandra? This could be especially important also when authors broach the question of the proportion of specialized individuals in both species. With this in mind, I think nevertheless that these findings seem to not be in contradiction with the expected restriction niche in a context of interspecific competition for S. salamandra which seems to focused on mainly 3 taxa. Even if it is difficult to compared with the few other studies elsewhere especially when they use a different technic (i.e. DNA), I feel that these previous works cited showed more diversified diets. On the other hand, since the trophic niche of S. italicus has only been studied in some caves, could we suspect here a broadening of the niche for this species? Some studies seem to indicate that niche widths can increased with intensified interspecific competition (i.e. doi: 10.1111/oik.08375) or more classically on the contrary when interspecific competition decrease. In this study, considering the high difference in density between the two species, authors underlined wisely that intraspecific competition among S. italicus may play a major role rather than interspecific competition.

Finally, I found that the following section in the conclusions is not very easy:

Line 310-313: “The prey consumed by both species mostly differed in size, making morphological constrains a potential tool useful to reduce competition. According to our results, physiological, morphological and behavioral constraints might play an important role in reducing the competition between sympatric generalist predators.”

Also resumed in the abstract line 24-25: “but specific physiological, morphological and behavioural traits worked as natural deterrent”.

And can be perhaps improved. When I related to the following paragraph :

Line 280-284

“This also supports the hypothesis that S. italicus rather chooses to do not prey on this type of prey, as it does not seem that the surveyed area is characterized by a particular low abundance of worm-like prey. Although we lack specific data to show, is noteworthy to mention that Pulmonata and Haplotaxida consumed by S. salamandra had an impressive size, certainly unsuitable for S. italicus.”

I can understand the arguments in favour of the hypothesis of a finally little overlaps of the basal trophic niches, S. Salamandra frequently consuming them in particular the biggest whereas S. italicus selected few frequently such preys and it this case are able to snap only smaller ones. But involving morphological, physiological or behavioural constraints or traits in this selection does not seem really appropriate.

I note also that these two taxa (Pulmonata and haplotaxida) are not systematically present in the diet of S. salamandra in other studies. Authors wrote lines 253-2553: “Indeed, this type of prey consumed by S. salamandra accounted for 79% of its diet (Table 1), corroborating similar results obtained from different populations throughout its distribution [13,38,49,73,74]. I suspect more a strong local variability among the worm-like preys.

Round 2

Reviewer 2 Report

The authors addressed my comments.